# Breastfeeding and Overweight in European Preschoolers: The ToyBox Study

**DOI:** 10.3390/nu13082880

**Published:** 2021-08-21

**Authors:** Natalya Usheva, Mina Lateva, Sonya Galcheva, Berthold V. Koletzko, Greet Cardon, Marieke De Craemer, Odysseas Androutsos, Aneta Kotowska, Piotr Socha, Luis A. Moreno, Yannis Manios, Violeta Iotova

**Affiliations:** 1Department of Social Medicine and Health Care Organization, Medical University of Varna, 9002 Varna, Bulgaria; 2Department of Pediatrics, Medical University of Varna, 9002 Varna, Bulgaria; mina_pl@yahoo.com (M.L.); sonya_galcheva@mail.bg (S.G.); iotova_v@yahoo.com (V.I.); 3Division of Metabolic and Nutritional Medicine, Department Paediatrics, Dr. von Hauner Children’s Hospital, LMU University Hospitals, 80337 Munich, Germany; berthold.koletzko@med.uni-muenchen.de; 4Department of Movement and Sports Sciences, Ghent University, 9000 Ghent, Belgium; greet.cardon@ugent.be; 5Department of Rehabilitation Sciences, Ghent University, 9000 Ghent, Belgium; marieke.decraemer@ugent.be; 6Research Foundation Flanders, 1000 Brussels, Belgium; 7Department of Nutrition and Dietetics, School of Physical Education, Sport Science and Dietetics, University of Thessaly, 382 21 Volos, Greece; oandrou@hua.gr; 8Public Health Department, Children’s Memorial Health Institute, 04-730 Warsaw, Poland; a.kotowska@ipczd.pl (A.K.); p.socha@ipczd.pl (P.S.); 9GENUD (Growth, Exercise, Drinking Behaviour and Development) Research Group, University of Zaragoza, 50009 Zaragoza, Spain; lmoreno@unizar.es; 10Department of Nutrition and Dietetics, Harokopio University, 176 76 Athens, Greece; manios@hua.gr

**Keywords:** breastfeeding, preschoolers, overweight, obesity

## Abstract

The benefits of breastfeeding (BF) include risk reduction of later overweight and obesity. We aimed to analyse the association between breastfeeding practices and overweight/obesity among preschool children participating in the ToyBox study. Data from children in the six countries, participating in the ToyBox-study (Belgium, Bulgaria, Germany, Greece, Poland, and Spain) 7554 children/families and their age is 3.5–5.5 years, 51.9% were boys collected cross-sectionally in 2012. The questionnaires included parents’ self-reported data on their weight, height, socio-demographic status, and infant feeding practices. Measurements of preschool children’s weight and height were done by trained researchers using standard protocols and equipment. The ever breastfeeding rate in the total sample was 85.0% (*n* = 5777). Only 6.3% (*n* = 428) of the children from the general sample were exclusively breastfed (EBF) for the duration of the first six months. EBF for four to six months was significantly (*p* < 0.001) less likely among mothers with formal education < 12 years (adjusted Odds Ratio (OR) = 0.61; 95% Confidence interval (CI) 0.44–0.85), smoking throughout pregnancy (adjusted OR = 0.39; 95% CI 0.24–0.62), overweight before pregnancy (adjusted OR = 0.67; 95%CI 0.47–0.95) and ≤25 years old. The median duration of any breastfeeding was five months. The prevalence of exclusive formula feeding during the first five months in the general sample was about 12% (*n* = 830). The prevalence of overweight and obesity at preschool age was 8.0% (*n* = 542) and 2.8% (*n* = 190), respectively. The study did not identify any significant association between breastfeeding practices and obesity in childhood when adjusted for relevant confounding factors (*p* > 0.05). It is likely that sociodemographic and lifestyle factors associated with breastfeeding practices may have an impact on childhood obesity. The identified lower than desirable rates and duration of breastfeeding practices should prompt enhanced efforts for effective promotion, protection, and support of breastfeeding across Europe, and in particular in regions with low BF rates.

## 1. Introduction

Optimal nutrition during the first years of life is important to promote healthy growth and development of children. Inappropriate nutrition augments the risk of illness and childhood obesity. Obesity is an escalating public health problem worldwide, with an expected prevalence of 9.1% among preschoolers in 2020 [1,2] and an estimated 70 million overweight or obese children and adolescents by 2025, the majority living in low- and middle-income countries [3]. According to the criteria of the International Obesity Task Force (IOTF), about 17.9% of European children aged 2–7 years were identified with overweight or obesity in the recently published systematic review, including 32 studies (*n* = 197,755 children) from 27 European countries [4]. Currently available options for treating overweight and obese children are less than satisfactory, therefore implementation of effective preventive measures, including promotion of optimal feeding from early on, is of utmost importance [5].

Breastfeeding is considered as a “golden standard”, the best nutrition for infants, and is associated with numerous short- and long-term health benefits, such as higher cognitive development scores and reduced risk of gastro-intestinal infections, otitis media, atopic dermatitis, and asthma, as well as later non-communicable diseases in adults such as type 2 diabetes and obesity [6,7,8,9,10,11]. It has been proposed that about one million cases of childhood obesity per year may be prevented by recommended breastfeeding (BF) practices [12]. The reported advantages of exclusive breastfeeding (EBF) compared to partial breastfeeding have led to the World Health Organization (WHO) global public health recommending exclusive breastfeeding for six months and continued breastfeeding for ≥two years with complementary foods introduced [13]. The protective role of BF against overweight and obesity was first reported in a large cross-sectional study in 1999 [14] and has since been replicated in many studies around the world [11,15,16,17]. Potential mechanisms underpinning this association are differences in the composition of human milk and breast milk substitutes and their impact on infant hormonal and metabolic response and growth characteristics. Behavioural differences and associated confounders such as mother’s body mass index (BMI), education, and tobacco use in pregnancy have also been discussed [18,19,20,21,22,23,24]. We analysed the association between breastfeeding and obesity among school children, with adjustment for important potential confounders relevant to overweight and obesity among the large European sample of schoolchildren participating in the ToyBox Study.

## 2. Materials and Methods

The ToyBox study (Available online: www.toybox-study.eu (accessed on 21 June 2021) adhered to the Declaration of Helsinki and the conventions of the Council of Europe on human rights and biomedicine. All participating countries obtained ethical clearance from the respective ethical committees and local authorities, and all parents/caregivers provided a signed consent form before being enrolled in the study.

Data was collected between May and June 2012 using the primary caregivers’ questionnaire, which was filled in by parents/caregivers of preschoolers in six European countries (Belgium, Bulgaria, Germany, Greece, Poland, and Spain). Parents/caregivers of preschoolers born between January 2007 and December 2008 who attended the randomly selected kindergartens within the provinces of Oost-Vlaanderen and West-Vlaanderen (Belgium), Varna (Bulgaria), Bavaria (Germany), Attica (Greece), Mazowieckie (Poland), and Zaragoza (Spain) were grouped in three socioeconomic levels and invited to participate in ToyBox study. The necessary information about the ToyBox study was presented previously [25]. A standardized self-administered questionnaire for primary caregivers’ was used to determine the perinatal data of preschoolers (comprising of anthropometric measurements at birth and breastfeeding practices during the first year of life), as well as participating families’ socio-demographic characteristics. Questions about the nutrition of children during the first 12 months of life were targeted on identifying the presence/absence of breastfeeding each month after birth and when water, tea, juice, formula milk, and solid/semi-solid foods were included. In order to minimize recall bias, the guidance to the parents/caregivers was to refer to the children’s medical records for the questions in the perinatal section. As a result, the test–retest reliability study showed an excellent value of ICC (intra-class correlation coefficient)—0.75 for this section of the questionnaire. The socio-demographic characteristics were previously found to have very high reliability (i.e., all ICCs ≥ 0.937), while the reliability of the questions on parental weight and height spread from moderate to very high (i.e., ICC ranged from 0.489 to 0.911) [26]. Other papers [27,28] presented the data collected using validated questionnaires as well as the corresponding results about health-related behaviors (dietary habits, physical activity, and sedentary behavior) of preschoolers and their parents.

Mothers’ years of education were used as stratification criteria for the socioeconomic status (SES) of the families by categorizing it as low (≤12 years), medium (13–16 years), and high (≥16 years of education).

Preterm birth was defined as <37 gestational weeks, and full-term birth as ≥37 gestational weeks. Questions on breastfeeding status were based on definitions according to the WHO indicators [29]. Exclusive breastfeeding was defined as BF with no other food or liquid given, except drops and syrups (vitamins, minerals, medicines). Breastfeeding with an additional supply of water or water-based liquids, such as fruit juice, tea, or syrups, was considered predominant breastfeeding. Full BF refers to either exclusive or predominant breastfeeding. The inclusion of other milk and foods (formula milk, and/or semi-solids) was considered partial breastfeeding. Ever breastfeeding rate is the proportion of infants less than 12 months who were ever breastfed. The “later stage BF initiation” was defined as starting of BF at any time but not from birth.

According to the study protocol, the questionnaires with more than 75% of the required information provided were included in the statistical analysis. The analysis did not include the children for which the information about feeding in the first two months (*n* = 444) nor about the time of solid foods’ introduction (*n* = 300).

### 2.1. Anthropometric Data

Children’s weight (0.1 kg measurement resolution) and height (0.1 cm measurement resolution) were obtained by means of a standardized protocol and equipment, subjected to calibration before and during the data collection period [30]. For the purpose of ensuring a very good intra- and inter- observed reliability agreement [31], research assistants, who passed extensive training before commencing the study, carried out all measurements to achieve. The definition of overweight, including obesity was based on the WHO criteria: for children age < 5 years as BMI z-score > 2 standard deviation (SD) and BMI z-score > 3 SD, respectively, whereas, for children age > 5 years, the ranges were encompassed into BMI z-score > 1 SD for overweight and into BMI z-score > 2 SD for obesity. Calculated ponderal index (PI = weight/height^3^) served for evaluation of children’s weight status at their birth with the normal PI range being 2.2–3.0 g/cm^3^, PI > 3.0 was considered indicative for overweight; while PI < 2.2 indicated low weight newborns. Parents/caregivers self-reported parental weight and height, while their BMI was calculated. Categorization of parents/caregivers with regard to their BMI defined the groups of normal weight (≤24.9 kg/m^2^), overweight (≥25 and ≤29.9 kg/m^2^), and obese (≥30 kg/m^2^) ones [32].

### 2.2. Statistical Analyses

Continuous variables are shown as the means ± standard deviation for the cases of normal distribution (e.g., age of preschoolers, age of mothers, the introduction of solid foods) and as the medians and IQR (interquartile range) for variables deviating from a normal distribution (duration of breastfeeding). Shapiro-Wilk tests were used for testing of normality of variables’ distribution.

With regard to country and children’s BMI categories, the χ^2^ test and Fisher’s exact test were applied to these categorical variables. For comparison of the means from two samples, an independent samples *t*-test was implemented, whereas for the means of more than two samples (birth weight; mother’s age)—the one-way ANOVA (analysis of variance) with a Scheffe’s post-hoc analysis.

The median Mann-Whitney test was used for the comparison of the medians from independent samples. Pearson’s and Spearman’s correlation analyses were utilized for exploration of the relationship between feeding practices and children’s BMI as well as mother’s characteristics. The odds of being overweight/obese (dependent variable), while accounting for different breastfeeding practices were estimated by means of logistic regression analysis with 95% confidence intervals (CIs). The validity of significant associations to the duration of breastfeeding, when adjusted for these other variables, potential confounding variables and covariates (mother’s age and BMI before pregnancy, SES (socio-economic status), smoking habits during pregnancy, and country) was determined on the basis of the obtained results. For the purpose of quantifying the probability of conforming to the current recommendations for the EBF in 4–6 months of age, logistic regression analysis was carried out adjusting for mothers’ age and BMI before pregnancy, SES, smoking habits during pregnancy, and country. Conformance to recommendations for EBF 4–6 months of age (yes/no) and overweight/obesity at preschool age (yes/no) were specified as dependent variables in the relevant models. The variables presented as logistic regression model coefficients were chosen on the basis of their relevance for the investigated subject and also these tested negatively for the presence of collinearity, thus not bringing any tangible influence too. The Statistical Package for Social Sciences (IBM SPSS v. 20, Chicago, IL, USA) was used for the data analysis with *p* < 0.05 set as a level of significance.

## 3. Results

A total of 6800 questionnaires from the six countries met the qualifying criteria for inclusion in the analysis is. The sociodemographic characteristics of respondents are shown in Table 1. The mean age of children is 4.75 ± 0.43 years; 52.3% boys, with no statistically significant difference in gender distribution between the participating countries. Further details with regard to characteristics of the ToyBox study sample are presented in other papers [25,32].

### 3.1. Breastfeeding Practices

The ever breastfeeding rate in the total sample is 85.0% (*n* = 5777). The highest rates are reported in the samples from Poland (94.7%) and Bulgaria (92.8%), while the lowest is in the Belgian sample (66.7%). Prevalence of the BF initiation from the time of birth is 96.1% (*n* = 5531) and has a statistically positive weak correlation with the country (Pearson’s *r* = 0.13; <0.001): the highest are in Belgium (99.6%) and Germany (99.1%). Children in Spain and Greece more often than in other study countries started BF not from birth but from a later time: 12.4%; *n* = 75 and 7%; *n* = 99, respectively.

Only 6.3% (*n* = 428) of the children were exclusively breastfed for the duration of the first six months after birth. Frequencies higher than the average of the total sample are observed for Germany (*n* = 163; 14.8%) and Poland (*n* = 105; 7.9%). Greece has the lowest frequency (2.7%); followed by Belgium (2.8%) and Spain (5.2%) (Table 2).

### 3.2. Duration of Breastfeeding

The median duration of BF among infants during the first 12 months is highest among Polish (9 months) and German (7 months) participants, with a significantly higher duration than the median duration of the entire sample (5 months, *p* < 0.001). For the entire sample, after adjustment for the risk factors there is a weak but significant positive relationship between the duration of breastfeeding and maternal characteristics: education level (Spearman’s *ρ* = 0.15; *p* = 0.023), SES (Spearman’s *ρ* = 0.09; *p* < 0.001), age at the pregnancy (Pearson’s *r* = 0.061; *p* < 0.0001). Additionally, mothers with a pre-pregnancy BMI ≥ 25(Pearson’s *r* = −0.14; *p* < 0.001) and smoking habits during pregnancy (Pearson’s *r* = −0.06; *p* < 0.0001) had a shorter duration of BF. Infants who were breastfed exclusively for 4–6 months have a longer duration of any BF (Pearson’s *r* = 0.32; *p* < 0.0001). Factors associated with the continued BF (>12 months) are mother’s education ≥ 14 years (*p* < 0.0001), normal weight before pregnancy (*p* < 0.0001), non-smoking habits during pregnancy (*p* < 0.0001), 25–40 years age group (*p* = 0.001) and SES (*p* = 0.02).

Exclusive formula feeding during first 4–6 months is about 12% (*n* = 830), with the highest value observed in the Belgian sample (*n* = 253; 22.4%) and the lowest in the Polish sample (4.5%; *p* < 0.001), which is associated with the following mothers’ characteristics: education ≥ 14 years (*p* < 0.0001), non-smoking habits during pregnancy (*p* < 0.0001), higher age (*p* < 0.0001) and higher SES (*p* = 0.025); normal pre-pregnancy weight (*p* = 0.03).

Mothers’ education less than 12 years (*p* < 0.0001); BMI > 25 kg/m^2^ (*p* < 0.0001); tobacco use throughout pregnancy (*p* < 0.001) and low SES (*p* < 0.001) are associated with a higher frequency of formula feeding. The same risk factors (maternal education less than 12 years (OR = 0.61; 95%CI 0.44–0.85), smoking throughout pregnancy (OR = 0.39; 95% CI 0.24–0.62), overweight before pregnancy (OR = 0.67; 95%CI (0.47–0.95)) are associated with not achieving EBF for 4–6 months (*p* < 0.05), Table 3.

### 3.3. Breastfeeding Practices and Weight Status of the Preschoolers

The prevalence of overweight and obesity according to the WHO criteria was 8.0% (*n* = 542) and 2.8% (*n* = 190), respectively (Table 1). Infant feeding practices exhibited a different association with the prevalence of overweight and obesity at different stages of childhood. A lower percentage of children were obese in the EBF 4–6 months group than the formula-fed group (1.6% vs. 6.5%; *p* = 0.012) (Table 4). Factors negatively related to the prevalence of overweight and obesity in preschoolers after controlling for confounding factors are EBF throughout the first three months of life (Pearson’s *r* = −0.03; *p* = 0.01) and non-significantly—duration of any breastfeeding (Spearman’s *ρ* = −0.02; *p* = 0.12).

Formula feeding at 4–6 months is linked to a higher prevalence of overweight and obesity at 6 months of infancy (13.8%) and among preschoolers (13.5%), but the association was non-significant when adjusted for confounding factors (Spearman’s *ρ* = 0.02; *p* = 0.18). (Table 4)

Table 5 shows the output from the logistic regression analysis with reducing factors for overweight or obesity being identified: the odds to become overweight at preschool age among children who were BF for 4–6 months is 0.87 (OR = 0.87; 95%CI 0.62–1.21; *p* = 0.40); 0.64 for EBF 6 months. The introduction of solid foods after six months of EBF is related to 69% less risk for overweight at preschool age (*p* = 0.25) in comparison to the formula milk feeding when adjusted for the mother’s characteristics (age and BMI before pregnancy, smoking habits during pregnancy, country, SES) and gender of the children.

Regarding the association between other characteristics at birth and obesity, there is no correlation with preterm or term delivery (*p* = 0.89) and PI at birth (*p* = 0.36) in the total sample and the country level stratification (*p* > 0.05).

## 4. Discussion

The analyses of the collected data from 6 countries in Europe regarding the characteristics at birth, infant feeding practices, and risk of childhood obesity revealed that in nearly all the countries 85% of children were breastfed, with the notable exception of Belgium, where only 66.7% of children were breastfed. In spite of the solid evidence demonstrating major health benefits associated with BF, including risk reduction for overweight and obesity, breastfeeding remains still well below the global goal of 50% EBF at 2025 [33,34,35]. Our data represent the BF practices and related factors corresponding to the year 2012 but are in agreement with other recent survey analyses indicating less than satisfactory breastfeeding practices in many European countries [36]. In our sample, only 6.3% of children were exclusively breastfed during the 6th month of life and thus met the WHO recommendation. However, a recent survey performed by the WHO Regional Office for Europe and the European Society of Pediatric Gastroenterology, Hepatology and Nutrition revealed that more than 82% of European countries recommended the introduction of complementary feeding at the age of 4–5 months and hence exclusive breastfeeding for a shorter period than six months [37]. The rate of EBF in our study is lower than results for EBF at 4–6 months from European samples such as IDEFICS (45.5%) and COSI (lowest—10.5% for Spain), and from the World Health Statistics WHO—Reports [38,39]. The significant data discrepancy appears to be primarily due to the different approaches of the calculation methodologies, and potentially also due to some imprecision due to recall bias in our study. In the studies cited above, as well as in almost all the official data, the frequency of exclusive breastfeeding is presented according to the WHO “Indicators for assessing infant and young child feeding practices” as the prevalence of “exclusively breastfed for the first 6 months of life”. Thus, this indicator does not measure the proportion of children meeting the WHO recommendation for exclusive breastfeeding of all children until 6 months, as pointed out by Pullum [40]. The prevalence data in the above-mentioned studies are very close to the goals in the “WHO global nutrition targets 2025” and can lead to inadequate political decisions regarding protection, promotion, and support of exclusive breastfeeding.

Children who were exclusively BF throughout the first three months of life were less likely to become overweight at preschool age when adjusted for country, age and gender, mother’s pre-pregnancy age and BMI.

The odds to become overweight at preschool age among children who were BF for 4–6 months is 0.87 and 0.31 for the EBF and solids introduction at 4–6 months in comparison to the formula milk feeding, when adjusted for mother’s characteristics (age and BMI before pregnancy, smoking habits during pregnancy, country, SES and gender of children). Thus the effect size for breastfeeding effects on later obesity in our study, although non-significant, is in the same order of magnitude as found in reviews and meta-analyses [15,16,17,18]. Our results also show a protective role of any breastfeeding against obesity and overweight, with 13% less risk at any BF of 4–6 months, and 69% for overweight and 12% for obesity at EBF and solid foods introduction for six months, compared to exclusive formula feeding. These results agree with the reported 26% decrease of the odds of overweight or obesity with any BF in 113 studies [23]. Also, infants fed formula during the first 4–6 months have a higher prevalence of overweight and obesity in preschool age (Table 4). Possible mechanisms for this relationship may include the different macronutrient composition of breast milk and formula, in particular the lower protein supply with breast milk [18], and potentially the presence of bioactive substances like ghrelin, leptin, insulin-like growth factor-1, adiponectin in human milk but not in formula [41]. There is published evidence that feeding formula milk has an accelerating effect on infant weight, height, body fat, apparently mediated through high levels of protein (the “early protein hypothesis”) [18] and lower appetite control of bottle-fed infants [15,20,42,43]. Moreover, a recent study reported that formula feeding in the early life of infants small for their gestational age is related to prospective overweight in preschool age, but only among girls [44]. Breastfeeding also modulates the physiological development of the digestive tract [45] and intestinal colonization [46], which might contribute to risk reduction for obesity in later life [47]. However, our results also show significant confounding of the association of breastfeeding and later overweight by low SES that is linked to both less breastfeeding success and more overweight. A more detailed analysis of the relationship between SES and overweight/obesity prevalence in children participating in the ToyBox study was previously published [32]. In the current analysis, SES is included as a confounder for which the analysis has been adjusted.

One of the limitations of the current presentation is the cross-sectional study design which is not able to identify the cause-effect association between the sociodemographic characteristics and the prevalence of overweight/obesity among preschoolers. Particularly, the selection of children from kindergartens within only one province in each country is the next methodological limitation of the study, which does not allow to draw a conclusion at a national level for each of the countries based on the collected data. Another limiting factor stems from the parental self-reporting of weight, height, gestational weight gain, and infant’s birth weight, which despite the relatively close time for recollection might have introduced recall bias. Retrieving data about BF duration through mothers’ recall for the time period 3–4 years ago can be seen as another limitation. This mostly impacts the reporting of EBF, as there is evidence that more than two years after birth mothers tend to overestimate the duration of BF [48]. In line with previous studies, our analysis indicates that breastfeeding practices are associated with factors that are also related to obesity development, such as maternal education, SES, BMI, and infant birth weight [38,49,50,51]. The statistical power to detect protective effects of breastfeeding on later obesity is limited by the fact that we could not compare exclusive breastfeeding from birth with exclusive formula feeding from birth, which has been reported to have the most marked effect on later obesity risk [52]. Even though we adjusted the calculation of ORs for these factors, residual confounding cannot be excluded, hence the extent of the causal effect of breastfeeding itself on later risk of overweight and obesity is difficult to determine.

A large number of study participants, the involvement of participants from several European countries which adds a level of independent validation, as well as the harmonized methodology in data collection and obtaining measurements may be noted as merits of this study [26,32].

## 5. Conclusions

Our study found only a non-significant trend for reduced childhood overweight associated with breastfeeding when adjusting for relevant confounding factors. It is likely that sociodemographic and lifestyle factors associated with breastfeeding practices have an impact on childhood obesity. The findings of less than desirable breastfeeding rates and duration underline the need for enhanced protection, promotion, and support of breastfeeding throughout Europe. Particularly intensive efforts are necessary for populations with low breastfeeding rates and duration based on geographic region and other risk markers such as lower education and socioeconomic class, tobacco smoking, parental overweight and obesity [5,36]; short duration of maternity leave, psychological factors as maternal perceived stress and postpartum depression [53].

## Figures and Tables

**Table 1 nutrients-13-02880-t001:** Characteristics of participants by country.

	Country *n* (%)							
	Belgium	Bulgaria	Germany	Greece	Poland	Spain	Total	*p*
Child’s gender								
Male	600 (53.2)	438 (50.1)	577 (52.3)	841 (51.1)	707 (53.0)	392 (55.0)	3555 (52.3)	0.37 *
Female	528 (46.8)	436 (49.9)	527 (47.7)	806 (48.9)	627 (47.0)	321 (45.0)	3245 (47.7)	
	1128 (100)	874 (100)	1104 (100)	1647 (100)	1334 (100)	713 (100)	6800 (100)	
Birth weight mean (±SD)								
	3.34 (0.51)	3.26 (0.53)	3.32 (0.54)	3.14 (0.53)	3.44 (0.55)	3.32 (0.50)	3.29 (0.54)	<0.001 **
Ponderal Index at birth								
Low	121 (10.7)	142 (16.2)	28 (2.6)	339 (20.7)	754 (59.6)	95 (13.2)	1633 (24.3)	<0.001 *
Normal	914 (81.0)	690 (78.9)	883 (80.8)	1267 (77.2)	499 (39.5)	554 (77.7)	4806 (71.6)	
High	93 (8.3)	42 (4.9)	28 (2.6)	34 (2.1)	12 (0.9)	66 (9.1)	274 (4.1)	
BMI at 6th month								
Under-/normal	788 (94.9)	445 (90.6)	917 (91.7)	1334 (92.8)	818 (89.1)	545 (92.1)	4897 (92.0)	<0.001 *
Overweight	35 (4.2)	21 (4.3)	65 (6.5)	85 (5.9)	75 (8.2)	41 (6.9)	322 (6.1)	
Obese	7 (0.9)	25 (5.1)	18 (1.8)	17 (1.3)	25 (2.7)	6 (1.0)	98 (1.9)	
BMI at 12th month								
Under-/normal	607 (94.4)	400 (82.1)	902 (91.0)	1231 (88.2)	749 (82.6)	515 (88.9)	4404 (88.0)	<0.001 *
Overweight	27 (4.2)	60 (12.4)	62 (6.3)	130 (9.3)	131 (14.4)	55 (9.5)	465 (9.3)	
Obese	9 (1.4)	27 (5.5)	27 (2.7)	35 (2.5)	27 (3.0)	9 (1.6)	134 (2.7)	
BMI categories of preschools *n* (%)								
Underweight	8 (0.7)	5 (0.6)	4 (0.4)	11 (0.7)	7 (0.5)	2 (0.3)	37 (0.5)	<0.001 *
Normal weight	1059 (93.9)	764 (87.4)	1024 (92.8)	1356 (82.3)	1214 (91.0)	613 (86.0)	6030 (87.8)	
Overweight	47 (4.2)	76 (8.7)	61 (5.5)	200 (12.1)	83 (6.2)	75 (10.5)	542 (8.0)	
Obese	14 (1.2)	29 (3.3)	14 (1.3)	80 (4.9)	30 (2.2)	23 (3.2)	190 (2.8)	
SES—*n* (%)								
Low SES	453 (40.2)	124 (14.2)	243 (22.0)	790 (48.0)	445 (33.4)	290 (40.7)	2345 (34.5)	<0.001 *
Medium SES	341 (30.2)	300 (34.3)	388 (35.1)	448 (27.2)	395 (29.6)	256 (35.9)	2128 (31.3)	
High SES	334 (29.6)	450 (51.5)	473 (42.8)	409 (24.8)	494 (37.0)	167 (23.4)	2327 (34.2)	
	1128 (100)	874 (100)	1104 (100)	1647 (100)	1334 (100)	713 (100)	6800 (100)	
Mother’s age—mean (±SD)								
	33.7 (4.7)	33.9 (4.4)	35.7 (5.1)	37.1 (4.4)	34.5 (4.3)	37.7 (4.6)	35.4 (4.7)	<0.001 **
BMI categories *n* (%)								
Under-/normal	755 (70.4)	667 (78.9)	727 (70.9)	1101 (70.1)	1011 (78.6)	503 (74.2)	4764 (73.5)	<0.001 *
Overweight	217 (20.2)	133 (15.7)	213 (20.8)	328 (20.9)	204 (15.9)	134 (19.8)	1229 (19.0)	
Obese	100 (9.3)	45 (5.3)	86 (8.4)	142 (9.0)	72 (5.6)	41 (6.0)	486 (7.5)	
Tobacco use during pregnancy								
No smoking	1011 (90.7)	687 (79.8)	956 (89.2)	1340 (82.7)	1220 (93.2)	574 (81.0)	5788 (86.6)	
Smoking 2nd trimester	2 (0.2)	12 (1.4)	4 (0.4)	40 (2.5)	1 (0.1)	2 (0.3)	61 (0.9)	<0.0001 *
Smoking at 1st & 3rd trimester	6 (0.5)	46 (5.3)	29 (2.7)	54 (3.3)	29 (2.2)	25 (3.5)	189 (2.8)	
Smoking throughout pregnancy	96 (8.6)	116 (13.5)	83 (7.7)	187 (11.5)	59 (4.5)	108 (15.2)	649 (9.7)	

* χ^2^ test; ** ANOVA; BMI—Body mass index; SES—socio-economic study; SD—standard deviation.

**Table 2 nutrients-13-02880-t002:** Breastfeeding practices among preschool children from the six countries, participating in the ToyBox study.

Infant Feeding Practice	Country—*n* (%)							*p*
	Belgium	Bulgaria	Germany	Greece	Poland	Spain	Total	
BF initiation—*n* (%)								
-from birth	727 (99.6)	798 (98.4)	920 (99.1)	1319 (93.0)	1236 (97.9)	531 (87.6)	5531 (96.1)	<0.001 ^†^
-from later stage	3 (0.4)	13 (1.6)	8 (0.9)	99 (7.0)	27 (2.1)	75 (12.4)	225 (3.9)	
-total	730 (100)	811 (100)	928 (100)	1418 (100)	1263 (100)	606 (100)	5756 (100)	
EBF 4–6 months—*n* (%)	32 (2.8)	47 (5.4)	163 (14.8)	44 (2.7)	105 (7.9)	37 (5.2)	428 (6.3)	<0.001 ^†^
Duration of BF (months; median; IQR)	4 (2–6)	5 (3–9)	7 (4–11)	3 (2–6)	9 (4–13)	5 (2–9)	5 (2–9)	<0.001 **
Continued BF rate (>12 months)	31 (3.9)	87 (10.3)	134 (12.1)	84 (5.9)	347 (26.3)	95 (15.7)	778 (12.8)	<0.001 ^†^
Introduction of SF months (mean; ±SD)	4.6 ± 1.8	6.6 ± 2.0	6.3 ± 1.8	5.8 ± 1.2 *	5.8 ± 1.6 *	5.6 ± 1.5 *	5.8 ± 1.7 *	<0.001
EBF 4–6 months + introduction SF and BF < 12 months	32 (2.84)	47 (5.38)	163 (14.76)	44 (2.67)	105 (7.87)	37 (5.19)	428 (6.29)	<0.001 ^†^
EBF 4–6 months + introduction SF and BF ≥ 12 months	15 (1.15)	24 (2.90)	74 (7.86)	18 (1.09)	142 (10.64)	18 (2.52)	225 (3.31)	<0.001 ^†^

* Introduction of solid foods is significantly different with exception of the next comparisons—Greece and Spain *p* = 0.13; Greece and Poland *p* = 0.9; Poland and Spain *p* = 0.18 (ANOVA; Scheffe Post-hoc test); ** *p*-value in median test; ^†^
*p*-value in χ^2^-test; BF—breastfeeding; EBF—exclusive breastfeeding; SF—solid foods; IQR—Interquartile range; SD—standard deviation.

**Table 3 nutrients-13-02880-t003:** Maternal characteristics associated with non-compliance to recommendations for exclusive breastfeeding (EBF) for 4–6 months.

	EBF 4–6 Months (*n* = 5975)	
	OR (CI 95%)	*p*
Education ^1^		
≤12 years	0.61 (0.44–0.85)	0.003
13–16 years	0.65 (0.52–0.83)	<0.0001
≥16 years	1 (reference)	
Smoking habits during pregnancy ^2^		
No smoking	1 (reference)	
Smoking during pregnancy	0.39 (0.24–0.62)	<0.0001
Age mother at pregnancy ^3^		
≤25 years	1 (reference)	
26–39 years	1.76 (1.15–2.70)	0.001
≥ 40 years	1.95 (0.93–4.12)	0.08
BMI before pregnancy ^4^		
Underweight	0.98 (0.68–1.41)	0.91
Normal-weight	1 (reference)	
Overweight	0.67 (0.47–0.95)	0.03
Obese	0.53 (0.27–1.05)	0.07

^1^ Adjusted for age and BMI before pregnancy, smoking habits during pregnancy, country and SES. ^2^ Adjusted for age and BMI before pregnancy, country and SES. ^3^ Adjusted for age BMI before pregnancy, smoking habits during pregnancy, country and SES. ^4^ Adjusted for age before pregnancy, smoking habits during pregnancy, country and SES; (BMI—Body mass index; SES—socio-economic status; EBF—exclusive breastfeeding)

**Table 4 nutrients-13-02880-t004:** Breastfeeding practicies and weight status of children.

	Weight 6th Month			Weight 12th Month			Weight Preschoolers		
	Under/Normal Weight	Over-Weight	Obese	Under/Normal Weight	Over-Weight	Obese	Under/Normal Weight	Over-Weight	Obese
EBF 0–3 months	1672 (91.7)	116 (6.4)	35 (1.9)	1557 (88.6)	163 (9.3)	37 (2.1)	2104 (90.8)	170 (7.3)	43 (1.9)
EBF 4–6 months	304 (90.2)	23 (6.8)	10 (3.0)	299 (88.7)	29 (8.6)	9 (2.7)	392 (91.6)	29 (6.8)	7 (1.6)
EBF ≥ 7 months	20 (83.3)	4 (16.7)	0	19 (79.2)	4 (16.7)	1 (4.2)	33 (94.3)	1 (2.9)	1 (2.9)
Predominant BF 0–3 months	852 (92.1)	55 (5.9)	18 (1.9)	777 (86.2)	95 (10.5)	29 (3.2)	1103 (90.2)	79 (6.5)	41 (3.4)
Predominant BF 4–6 months	222 (91.0)	15 (6.1)	7 (2.9)	207 (87.3)	20 (8.4)	10 (4.2)	295 (89.9)	28 (8.5)	5 (1.5)
Formula feeding 4–6 months	143 (86.1)	12 (7.2)	11 (6.6)	137 (83.0)	21 (12.7)	7 (4.2)	199 (86.5)	16 (7.0)	15 (6.5)
Duration of BF >12 months	1446 (91.3)	105 (6.6)	32 (2.0)	1315 (87.3)	155 (10.3)	36 (2.4)	1869 (89.8)	165 (7.9)	47 (2.3)

BF—breastfeeding; EBF—exclusive breastfeeding.

**Table 5 nutrients-13-02880-t005:** Logistic regression analysis of the association of infant feeding practices and the overweight or obesity among preschool children.

Infant Feeding Practice ^1^	Overweight		Obesity	
	OR (CI 95%)	*p*	OR (CI 95%)	*p*
Breastfeeding				
1–3 months	1.09 (0.81–1.45)	0.59	1.15 (0.69–1.93)	0.59
4–6 months	0.87 (0.62–1.21)	0.40	0.89 (0.50–1.60)	0.75
7–12 months	1.01 (0.73–1.41)	0.96	1.25 (0.69–2.26)	0.44
≥12 months	1.02 (0.70–1.50)	0.91	1.11 (0.54–2.25)	0.78
Formula feeding	1 (reference)		1 (reference)	
Infant feeding practicies for 6 months				
EBF	0.64 (0.22–1.81)	0.40	1.10 (0.63–1.93)	0.73
Predominant BF	1.12 (0.78–1.61)	0.55	0.91 (0.52–1.60)	0.74
Partial BF	0.95 (0.71–1.28)	0.73	1.35 (0.83–2.20)	0.23
EBF and solids	0.31 (0.04–2.30)	0.25	0.88 (0.47–1.77)	0.79
Formula feeding	1 (reference)		1 (reference)	

^1^ Adjusted for mother’s age and BMI before pregnancy, smoking habits during pregnancy, country, SES and children’s gender; EBF—exclusive breastfeeding; BF—breastfeeding.

## Data Availability

The data presented in this study are available on request from the corresponding author. The data are not publicly available due to restrictions of informed consent and the requirement of IRB review and approval.

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
