# Peer review of "Breastfeeding and Overweight in European Preschoolers: The ToyBox Study"

_nutrients, 2021, doi:10.3390/nu13082880_

Round 1

Reviewer 1 Report

The manuscript title “Breastfeeding and overweight in European preschoolers: the ToyBox study” by  Natalya Usheva et collegues is an interesting and well-conceived and clearly written manuscript on an important topic to understand if in effect the metabolic relationships and reciprocal influences between mother and infant during lactation could determine metabolic changing later in life.  Please see also this study Integrative Journal of Nursing and Health, 1(1): 35-41 (2018).

Author Response

The manuscript title “Breastfeeding and overweight in European preschoolers: the ToyBox study” by  Natalya Usheva et collegues is an interesting and well-conceived and clearly written manuscript on an important topic to understand if in effect the metabolic relationships and reciprocal influences between mother and infant during lactation could determine metabolic changing later in life.  Please see also this study Integrative Journal of Nursing and Health, 1(1): 35-41 (2018). 

We are very grateful for the Reviewer’s   positive evaluation and suggested study, which was cited in the manuscript (line 339).

Reviewer 2 Report

Title: Breastfeeding and overweight in European preschoolers: the ToyBox study

The study focused on the association between breastfeeding practices and the prevalence of overweight/obesity among preschoolers in six European countries. The article covers very important aspects related to breastfeeding as a potential factor in the prevention of childhood overweight and obesity. I have minor comments on the results of the study. Please address the following issues:

  • Data was collected between May and June 2012, which should be highlighted in the discussion of the results. The length and frequency of breastfeeding 10 years ago, as well as the factors influencing breastfeeding, may have varied in time.
  • Data presented in Table 1 should be more clearly divided into child and mother data
  • I have some doubts about the interpretation of data presented in tables 2 and 4. It is not always clear how the percentages were calculated and what (or why) constitutes 100%.
  • Breastfeeding practices categories presented in table 4 should be the same as Infant feeding practice provided in table 5.
  • In the lines 326-327 the authors state: “Strengths of this study are the large number of study participants…”. However, in my opinion the number of study participants, and particularly selection of children from kindergartens within only one province in each country is one of the limitations of the presented report. It should be highlighted that data is not nationally representative. Therefore, the results of the study should be interpreted with caution as not being representative of individual countries.
  • In addition, some literature items require citation corrections (e.g., 1, 3, 36).

In conclusion, the article requires only minor revisions before being published.

Author Response

The study focused on the association between breastfeeding practices and the prevalence of overweight/obesity among preschoolers in six European countries. The article covers very important aspects related to breastfeeding as a potential factor in the prevention of childhood overweight and obesity. I have minor comments on the results of the study. Please address the following issues:

We are very grateful for the Reviewer’s in-depth comments and positive evaluation.

  • Data was collected between May and June 2012, which should be highlighted in the discussion of the results. The length and frequency of breastfeeding 10 years ago, as well as the factors influencing breastfeeding, may have varied in time.

We thank the Reviewer for this very accurate remark and the clarification will be described in the Discussion section  (line293- 294)

  • Data presented in Table 1 should be more clearly divided into child and mother data.

Table 1 will be divided into two parts, representing child and mother’s characteristics separately.

  • I have some doubts about the interpretation of data presented in tables 2 and 4. It is not always clear how the percentages were calculated and what (or why) constitutes 100%.

Percentages in the Table 2 will be calculated by columns and both total numbers and 100% will be presented additionally.

In Table 4 the percentages will be calculated by rows and weight status will be presented in age periods separately for different types of nutrition.

  • Breastfeeding practices categories presented in table 4 should be the same as Infant feeding practice provided in table 5.

In Table 5 we focused on the infant feeding practices for 6 months, due to the WHO recommendation for 6 months EBF and  the risk for overweight/obesity among different types of nutrition was presented for the same age.

  • In the lines 326-327 the authors state: “Strengths of this study are the large number of study participants…”.However, in my opinion the number of study participants, and particularly selection of children from kindergartens within only one province in each country is one of the limitations of the presented report. It should be highlighted that data is not nationally representative. Therefore, the results of the study should be interpreted with caution as not being representative of individual countries.

We highly appreciate this very accurate remark made by the Reviewer and the limitation will be included in manuscript.

  • In addition, some literature items require citation corrections (e.g., 1, 3, 36).

We are grateful to the Reviewer for noticing this technical mistake which will be corrected.

In conclusion, the article requires only minor revisions before being published.

Reviewer 3 Report

It was nice to read the Usheva et al article. It is interesting to know the % and BF duration and introduction in different countries in Europe. It needs to get current data, even there was no association between BF practices and preschooler weight.  In addition, would be important to know, that EBF is the gold nutrition standard for neonates, however, the association with these practices during the early infancy and later on of infant life could be masked by multiple other factors that need to be explored. In addition, I would like to suggest some comments in the article. 

General comments: There are multiple subjective words (i.e. "very good", "excellent value", "weak") that could suggest the readers. Please, make your text more objective. 

Introduction:

  • Exclusive BF practices need to be defined. in addition, in line 72, clarify continued BF for 2 years with complementary foods introduced.

Material and methods:

  • How the reliability were extracted?
  • The definition of SGA by ponderal index needs a cite. In addition, the categorization of overweight and obesity is not clear for me. Could you please re-write it? 
  • The % comparison (line 152) should be tested by chis-squared of Fisher exact test. In addition, what was the median test (line 154)? What kind of correlation analysis was used? What was the cut-off of collinearity to be introduced in the models? 

Results:

  • It would be more useful if the authors could talk about children's sample size and not questionnaires. 
  • Table 1. The posthoc test in the ANOVA test needs to be reported. In addition, these variables were tested by normality? Furthermore, in all tables, the authors should consider using Fisher's exact test in % comparisons. 
  • How "later stage" was defined? 
  • Table 4. There was any significant between weight and BF time? 

Discussion:

  • lines 255-261. It would be possible that the observed rates are related to the work-politics in each country. The authors could emphasize the idea of work-family conflicts which can impact maternal-infant health (PMID: 33007816; PMID: 33807903). 

Minor comments:

  • Abstract, these ORs were the adjusted OR (aOR)?
  • line 31, decimal is separated by points. 
  • line 80 "school children"
  • lines 134-136 are in blue. 
  • line 161, defined SES.
  • lines 283, the parenthesis information is not necessary because is mentioned in the results section. 

Author Response

It was nice to read the Usheva et al article. It is interesting to know the % and BF duration and introduction in different countries in Europe. It needs to get current data, even there was no association between BF practices and preschooler weight.  In addition, would be important to know, that EBF is the gold nutrition standard for neonates, however, the association with these practices during the early infancy and later on of infant life could be masked by multiple other factors that need to be explored. In addition, I would like to suggest some comments in the article. 

General comments: There are multiple subjective words (i.e. "very good", "excellent value", "weak") that could suggest the readers. Please, make your text more objective. 

We are very thankful for the Reviewer’s in-depth comments and positive evaluation.

Introduction:

  • Exclusive BF practices need to be defined. in addition, in line 72, clarify continued BF for 2 years with complementary foods introduced.

We are grateful to the Reviewer for the suggestion and with respect to it, the continued BF for 2 years is now clarified (lines 72-73). The definition of Exclusive BF is described in the Methodology section (lines 123-124).

Material and methods:

  • How the reliability were extracted?

Reference #27 (González-Gil, E.M., T. Mouratidou, G. Cardon, O. Androutsos, I. De Bourdeaudhuij, M. Góz´dz´, N. Usheva, J. Birn-baum, Y. Manios and L. A. Moreno on behalf of the ToyBox-study group, Reliability of primary caregivers reports on lifestyle behaviours of European pre-school children: the ToyBox-study. Obes Rev, 2014. 15(S3): p. 61-66.) describes the reliability analysis of the parents questionnaire:

“Parents/ caregivers from six countries (Belgium, Bulgaria, Germany, Greece, Poland and Spain) were asked to complete the questionnaire twice within a 2-week interval. A total of 93 questionnaires were collected. Test–retest reliability was assessed using intra-class correlation coefficient (ICC). Reliability of the six questionnaire sections was assessed. A stronger agreement was observed in the questions addressing sociodemographic and perinatal factors as opposed to questions addressing behaviours. “

  • The definition of SGA by ponderal index needs a cite. In addition, the categorization of overweight and obesity is not clear for me. Could you please re-write it? 

We highly appreciate this very accurate remark made by the Reviewer and the description of PI is now clarified (lines148-153).

  • The % comparison (line 152) should be tested by chis-squared of Fisher exact test.

In addition, what was the median test (line 154)?

The Mann-Whitney test was used for comparison of medians from independent samples.

  • What kind of correlation analysis was used? 

Pearson’s and Spearman’s correlation analyses were utilized for exploration of the relationship .

  • What was the cut-off of collinearity to be introduced in the models? 

Collinearity was measured by variance inflation factors (VIF) and the value 5 was used as a cut-off point for collinearity.

Results:

  • It would be more useful if the authors could talk about children's sample size and not questionnaires. 

We thank the Reviewer for this suggestion. The qualifying criteria for inclusion in the analysis according to the study protocol (presence of over 75% of the required information) was set with reference to the questionnaires and hence the number of questionnaires is mentioned instead of children’s sample size. 

  • Table 1. The posthoc test in the ANOVA test needs to be reported. In addition, these variables were tested by normality? Furthermore, in all tables, the authors should consider using Fisher's exact test in % comparisons. 

The ANOVA Scheffe Post-hoc test was applied and the different p-values than presented in the last column of the relevant tables are described in the legend under the same table.

  • How "later stage" was defined? 

The “later stage” was defined as starting of BF at any time but not from the birth.

  • Table 4. There was any significant between weight and BF time? 

This could be find in the text (lines 263-267) : Factors negatively related to the prevalence of overweight and obesity in preschoolers after controlling for confounding factors are EBF throughout the first 3 months of life (Pearson’s r=-0.03;p=0.01) and non-significantly - duration of any breastfeeding (Spearman’s ρ = -0.02; p=0.12).

Discussion:

  • lines 255-261. It would be possible that the observed rates are related to the work-politics in each country. The authors could emphasize the idea of work-family conflicts which can impact maternal-infant health (PMID: 33007816; PMID: 33807903). 

We value highly this comment made by the Reviewer and believe that the relationship between the BF frequency and practices, on one hand, and the labor legislation regarding maternity in different countries, on another, could be our aim in a future article.

Minor comments:

We thank the Reviewer for the careful review and the text is now updated with these comments.

  • Abstract, these ORs were the adjusted OR (aOR)?
  • line 31, decimal is separated by points. 
  • line 80 "school children"
  • lines 134-136 are in blue. 
  • line 161, defined SES.
  • lines 283, the parenthesis information is not necessary because is mentioned in the results section.